# ALICE: AN INTERPRETABLE NEURAL ARCHITECTURE FOR GENERALIZATION IN SUBSTITUTION CIPHERS

## ABSTRACT

We present cryptogram solving as an ideal testbed for studying neural network reasoning and generalization; models must decrypt text encoded with substitution ciphers, choosing from 26! possible mappings without explicit access to the cipher. We develop ALICE (an Architecture for Learning Interpretable Cryptogram dEcipherment), a simple encoder-only Transformer that sets a new state-of-the-art for both accuracy and speed on this decryption problem. Surprisingly, ALICE generalizes to unseen ciphers after training on only $\sim$1500 unique ciphers, a minute fraction ($3.7 \times 10^{-24}$) of the possible cipher space. To enhance interpretability, we introduce a novel bijective decoding head that explicitly models permutations via the Gumbel-Sinkhorn method, enabling direct extraction of learned cipher mappings. Through early exit and probing experiments, we reveal how ALICE progressively refines its predictions in a way that appears to mirror common human strategies—early layers place greater emphasis on letter frequencies, while later layers form word-level structures. Our architectural innovations and analysis methods are applicable beyond cryptograms and offer new insights into neural network generalization and interpretability. Code is released for reproducibility at https://anonymous.4open.science/r/alice-4D23/.

## 1 INTRODUCTION

A cryptogram is a type of puzzle in which text is encrypted using a substitution cipher, and the user's task is to recover the original plaintext by inferring the cipher used for the encryption. Users typically solve cryptograms based on prior knowledge about language letter frequency distributions and common words. Originally developed for real encryption purposes, they are now popular in newspapers and puzzle books for entertainment purposes due to their simplicity. This simplicity, however, provides a unique testbed for testing and understanding generalization and reasoning in neural networks.

### 1.1 CRYPTOGRAM TASK

In a one-to-one monoalphabetic substitution cipher, each letter in a fixed alphabet is mapped to a unique substitute character; this cipher represents a bijective mapping over the alphabet. While other ciphers exist (e.g., Vigenère cipher, Playfair cipher), we focus here on one-to-one monoalphabetic substitution ciphers, as the problem space is extremely large but remains structurally simple to interpret. We hereafter mean one-to-one monoalphabetic substitution cipher when we say "cipher", unless otherwise specified.

More formally, let $\Sigma$ be a finite alphabet of size $V$ representing allowable characters (e.g., 26 for the English alphabet). Then $S_V$ is the symmetric group on $\Sigma$ containing all bijections from $\Sigma$ to itself. Importantly, the cardinality of $S_V$ is $|S_V| = V!$. Each cipher mapping $f : \Sigma \to \Sigma$ is a permutation for $\Sigma$, with $f \in S_V$. Let $\mathbf{x} = (x_1, x_2, \dots, x_L) \in \Sigma^L$ be a plaintext sequence of characters (i.e., in English) and $\mathbf{c} = (c_1, c_2, \dots, c_L) \in \Sigma^L$, with $c_i = f(x_i), i \in \{1, \dots, L\}$ be the corresponding ciphertext sequence.

The goal is to develop a model that is trained on example pairs of $(\mathbf{c}, \mathbf{x})$, but is never given access to the underlying cipher $f$. At inference time, the model is tasked with decrypting some ciphertext

**c** that has been encrypted under an unseen cipher $f$. That is, the model needs to implicitly infer the inverse cipher mapping $f^{-1}$ and perform $\hat{\mathbf{x}} = f^{-1}(\mathbf{c})$.

This task is combinatorially complex and requires reasoning over a space with $V!$ possible ciphers. Because of the enormous number of possible $f$, a model that simply memorizes training examples will fail when applied to the decryption of unseen ciphers. Instead, a model that performs well must learn a general algorithm for decryption that will work under any cipher.

## 1.2 RELATED WORK

Cryptogram decryption is a long-studied problem, traditionally tackled through letter- and word-frequency analysis. These approaches typically require human domain knowledge, such as empirical frequency tables or dictionaries, to guide plaintext guesses (see Appendix A for a broader discussion). Neural approaches to substitution ciphers have only recently emerged, starting in 2018 and remaining relatively limited in scope. Gomez et al. (2018) applied a discrete GAN to Caesar and Vigenère ciphers without relying on prior frequency statistics or key information. Kambhatla et al. (2018) used an LSTM-based character language model with beam search (widths up to 100,000) to evaluate candidate plaintexts. Aldarrab & May (2021) introduced a multilingual encoder–decoder Transformer that first remaps ciphertext into a frequency-based representation before autoregressively decoding character by character. More recently, Kambhatla et al. (2023) proposed a recurrence encoding scheme with a decoder-only Transformer, extending the task to homophonic ciphers where a single plaintext character may map to multiple ciphertext symbols.

Despite these advances, existing neural methods generally lack interpretability and fail to enforce the bijective constraints fundamental to standard substitution ciphers. Large language models (LLMs) might seem like a natural alternative, but as we show in Appendix I, they often hallucinate and do not respect bijectivity, limiting their effectiveness for cryptogram decipherment.

## 1.3 MAIN CONTRIBUTIONS

We introduce ALICE, a simple encoder-only Transformer that achieves state-of-the-art accuracy and speed in cryptogram solving and serves as a useful instrument for studying neural generalization and reasoning. Our contributions are threefold:

**Architecture:** We develop a novel decoding head that explicitly enforces bijectivity. This architectural innovation is applicable to any domain with bijective mappings (e.g., matching problems, permutation learning) and provides a principled alternative to post-hoc constraint satisfaction or regularization.

**Interpretability:** ALICE enables direct extraction of learned permutations, eliminating the need for unreliable attention map analysis. Our early exit and probing analyses reveal interpretable decryption strategies that mirror human problem-solving approaches, providing a new perspective on how neural networks reason under structural constraints.

**Generalization:** We use ALICE to analyze neural network generalization in combinatorially complex domains, showing that robust performance emerges after exposure to only $3.7 \times 10^{-24}$ of the total task space. This has implications for few-shot learning in structured reasoning tasks.

We release all model and training code publicly at `https://anonymous.4open.science/r/alice-4D23/`.

## 2 MODEL ARCHITECTURE

ALICE (an Architecture for Learning Interpretable Cryptogram dEcipherments) is a family of encoder-only Transformers (Vaswani et al., 2017) designed for cryptogram solving. The backbone follows a LLaMA-style architecture (Touvron et al., 2023) with full quadratic attention, pre-normalization via RMSNorm (Zhang & Sennrich, 2019), SwiGLU activations (Shazeer, 2020), and Rotary Position Embeddings (Su et al., 2023). Panel (a) of Figure 1 illustrates the core architecture shared across all ALICE variants.

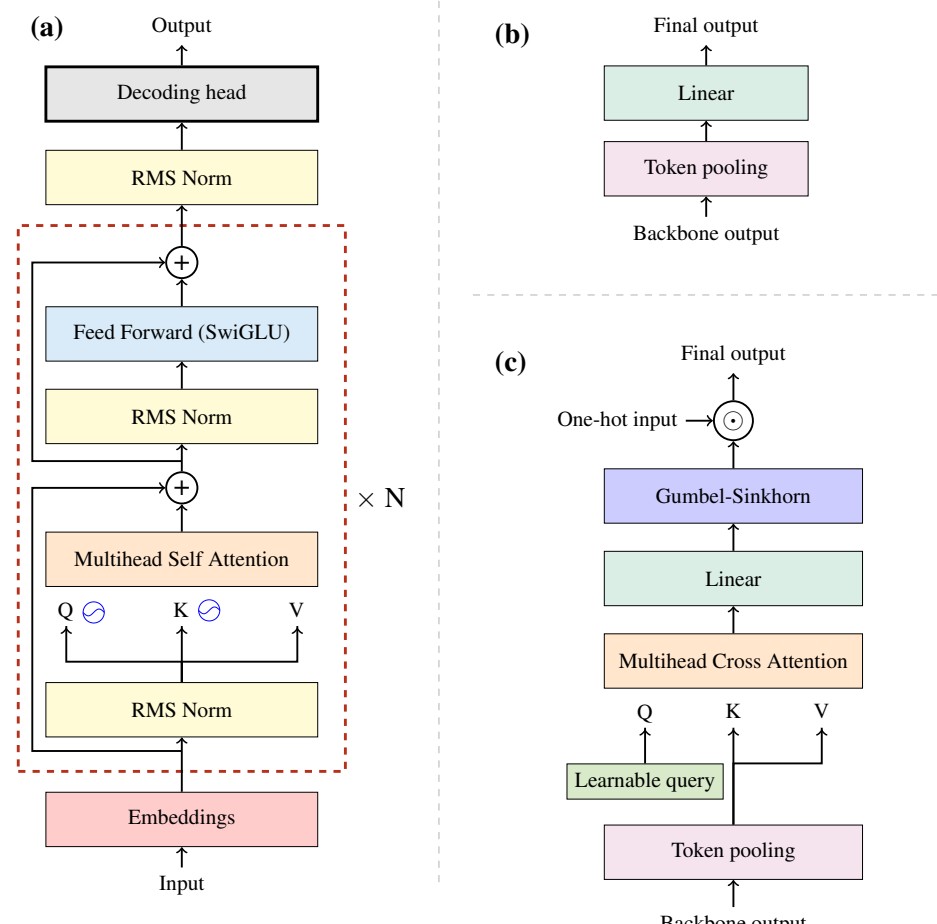

Figure 1: Overview of model architecture. **(a)** Main architecture and description of backbone. The ⊘ indicates Rotary Position Embedding (Su et al., 2023). The architectural details of the decoding head are shown on the right two panels. **(b)** Standard decoding head. "Backbone output" indicates the input to the decoding head; this is the representation produced by the encoder model after the final RMSNorm layer. **(c)** Bijective decoding head. "One-hot input" indicates the one-hot encoded version of the original ciphertext input. The ⊙ indicates a matrix multiplication operator. See Equation 5 for details on the Gumbel-Sinkhorn operation.

We study two models within this framework: ALICE-BASE, which uses a standard linear classification head to map hidden states to symbol predictions; ALICE-BIJECTIVE, which replaces the classification head with a novel end-to-end differentiable bijective decoding head, which we detail in Section 2.1.

**Symbol-wise token pooling strategy**    In the cryptogram task setup, each input letter is deterministically mapped to a single output letter. However, an "out-of-the box" standard neural network will not obey this constraint; the same input character appearing in different positions may produce different (i.e., inconsistent) outputs due to varying contextual embeddings. To address this problem, we implement a symbol-wise token pooling strategy. Prior to decoding from the final linear layer to output letters, we average all the embeddings for each unique symbol in the input (e.g., all instances of "A") and return a single pooled embedding for that symbol. This ensures that repeated symbols are always mapped consistently.

## 2.1 BIJECTIVE DECODING

Because we focus on one-to-one cipher mappings in this paper, we design a decoding module that allows ALICE to explicitly learn a bijective cipher mapping $f^{-1}$; we dub this variant ALICE-BIJECTIVE. This reduces hallucination and increases interpretability. To our knowledge, ALICE-BIJECTIVE is the first model explicitly designed with enforced bijectivity between input and output alphabets for solving cryptograms. While models without this bijective decoding module can implicitly learn a mapping $f^{-1}$, there are typically no constraints on its bijectivity, leading to the possibility of structurally invalid outputs (i.e., two ciphertext letters map to the same plaintext letter), and making it challenging to extract the learned mapping for interpretation.

ALICE-BASE will sometimes predict a mapping that maps two different letters to the same letter:

```
Plaintext:   THE SEA HAS TESTIFIED THAT AFRICA AND EUROPE HAVE KISSED.
Ciphertext:  JAE CEH AHC JECJISIEK JAHJ HSOIWH HQK EMOZUE AHDE YICCEK.
Predicted:   THE SEA HAS TESTIFIED THAT AFRICA AND EURUPE HAVE KISSED.
```

In this example, ALICE-BASE predicts that both the cipher letters "M" and "Z" map to the plaintext letter "U" due to a lack of bijective enforcement. We know, however, that under the cryptogram puzzle constraints, this should be impossible.

To mitigate this behavior, we add a new component to ALICE to learn a permutation matrix. Architecturally, we add a multi-head cross-attention layer with a learnable query after the main Transformer blocks to reduce the sequence length of the internal representation to the vocabulary size. We then apply a linear layer, as in the standard architecture, to reduce the model dimension to the vocabulary size. At training time, we then apply Sinkhorn normalization via iterative column- and row-wise normalization to turn this square matrix into a doubly stochastic matrix (Knopp & Sinkhorn, 1967):

$$S^0(X) = \exp(X) \tag{1}$$

$$S^\ell(X) = \mathcal{T}_c(\mathcal{T}_r(S^{\ell-1}(X))) \tag{2}$$

$$S(X) = \lim_{\ell \to \infty} S^\ell(X), \tag{3}$$

where $\mathcal{T}_c$ and $\mathcal{T}_r$ are column and row normalization (i.e., we divide each column value by the sum of its column and each row value by the sum of its row). From this doubly stochastic matrix, we would like to create a corresponding permutation matrix $\mathcal{P}^*$, typically framed as a linear assignment problem

$$\mathcal{P}^* = M(X) = \underset{P \in P_N}{\arg\max} \langle X, P \rangle_F, \tag{4}$$

where $P_N$ is the set of all permutation matrices and $\langle \cdot, \cdot \rangle_F$ is the Frobenius norm. However, as $M$ is not differentiable, we instead draw permutation matrices $\mathcal{P}$ from the Gumbel-Sinkhorn distribution (Mena et al., 2018)

$$\mathcal{P} \sim S\left(\frac{X + \epsilon}{\tau}\right), \text{ with } \epsilon \sim \text{Gumbel} \tag{5}$$

which is a continuous relaxation of $M$ that is fully differentiable. From this (soft) permutation matrix we can obtain soft decodings of the ciphertext by applying a matrix multiplication with the one-hot encoded inputs. Note that at inference time, we obtain a hard permutation matrix $\mathcal{P}^*$ using $M(X)$, as we do not need the output to be differentiable.

Panels (b) and (c) of Figure 1 show the differences between the ALICE-BASE decoding head and the ALICE-BIJECTIVE decoding head, respectively. ALICE-BASE only needs to learn the action of a mapping $f^{-1}(\cdot)$, and does not need to explicitly model the mapping itself. Thus, to recover the mapping for analysis, one must resort to strategies such as analysis of attention maps, which requires human intervention and can be time consuming as well as unreliable (Jain & Wallace, 2019; Serrano & Smith, 2019). ALICE-BIJECTIVE explicitly predicts a latent permutation matrix, and so extraction/inspection of the learned mappings can be done directly.

| Ciphertext length | $<32$ | $32-64$ | $64-128$ | $128-256$ | $>256$ |
|---|---|---|---|---|---|
| ALICE-BASE | $5.88^{+23.41}_{-5.88}$ | $0.00^{+4.88}_{-0.00}$ | $0.00^{+0.00}_{-0.00}$ | $0.00^{+0.00}_{-0.00}$ | $0.00^{+0.00}_{-0.00}$ |
| ALICE-DYNAMIC | $4.76^{+27.39}_{-4.76}$ | $0.00^{+4.88}_{-0.00}$ | $0.00^{+0.00}_{-0.00}$ | $0.00^{+0.00}_{-0.00}$ | $0.00^{+0.00}_{-0.00}$ |
| ALICE-BIJECTIVE | $7.14^{+31.91}_{-7.14}$ | $0.00^{+5.66}_{-0.00}$ | $0.00^{+0.89}_{-0.00}$ | $0.00^{+0.00}_{-0.00}$ | $0.00^{+0.00}_{-0.00}$ |

Table 1: Median SER (per-character error rate) for different test cipher lengths (lower is better). We compare the standard model (ALICE-BASE), a variant with dynamic embeddings (ALICE-DYNAMIC), and one with bijective decoding (ALICE-BIJECTIVE). Errorbars show the 16–84th percentile range across sequences within each length bin. Ciphertext length includes spaces and punctuation.

## 3 NUMERICAL EXPERIMENTS

**Dataset**   The main dataset we use is QUOTES500K (Goel et al., 2018), which consists of 500K English language quotes. We perform basic cleaning and keep sequences of lengths 15-300; further details on the dataset construction can be found in Appendix B. We take 97.5% of the sequences as training sequences and the remaining 2.5% as unseen testing sequences. Training examples are encrypted on-the-fly by applying a random substitution cipher to the plaintext, leaving spaces and punctuation unchanged.

**Performance evaluation**   Based on previous work (Nuhn et al., 2013; Kambhatla et al., 2018; Aldarrab & May, 2021; Kambhatla et al., 2023), we use the symbol error rate (SER) as our metric for evaluating the performance of our models; this is the fraction of incorrect characters in the output of the model as compared to the plaintext. We include the encrypted characters, spaces, and punctuation in the SER and report all SER values as percentages in this text.

### 3.1 ABLATIONS

To compare different design choices for ALICE, we perform two architecture ablations. The performance of all model variants is shown in Table 1. All models are roughly equal in terms of parameter count and are all trained for 200K steps with the same optimizer, hyperparameters, and dataset. Further details about training are provided in Appendix C.

**Dynamic embeddings**   Typically, embeddings are vector representations of tokens (letters). When trained on vast amounts of data, the meaning of tokens are learned, so that semantic information is encoded in the embeddings. In the case of cryptograms, while the input letters stay fixed, the *meaning* of a particular letter changes with the cipher. For example, "A" in ciphertext could either represent "E" or "S" in plaintext, depending on the cipher. This, as pointed out by Aldarrab & May (2021), violates the typical assumptions of embeddings. Thus, we want a way to allow for flexibility in the meaning of input tokens.

We experiment with *dynamic embeddings*: we create a hypernetwork to predict the token embeddings based on the input. We dub this variant ALICE-DYNAMIC and provide further architectural details in Appendix D. We hypothesize that this will allow for more flexibility than static embeddings (i.e., shared across all inputs/ciphers), which force "A" to have the same semantic embedding regardless of the cipher it is encoded with. In contrast, dynamic embeddings will allow the meaning of "A" to change with the context. However, we find in Table 1 that ALICE-DYNAMIC does not improve performance in any statistically significant way. Surprisingly, ALICE-BASE performs just as well for all ciphertext lengths, despite the meaning of tokens changing with each example. These results challenge prior work suggesting that one needs to carefully design a flexible embedding scheme in order to achieve strong performance (Aldarrab & May, 2021).

**Bijective decoding**   We use ALICE-BIJECTIVE as described in Section 2, which enforces the structure of the cryptograms task and improves interpretability. We find in Table 1 that ALICE-BIJECTIVE achieves performance that is slightly worse than ALICE-BASE, in particular for extremely short sequences ($<32$ characters) where performance difficulty is the highest, but this perfor-

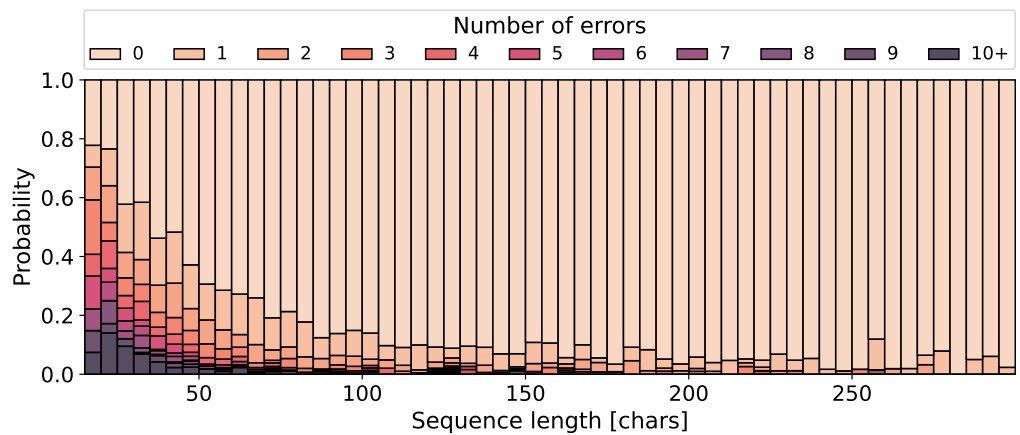

Figure 2: Distributions of number of errors made across various test ciphertext lengths for ALICE-BASE. The sequence length axis has a bin width of 5. Lighter colours indicate fewer errors.

| Ciphertext length $\longrightarrow$ | <**128** [%] | >**128** [%] |
|---|---|---|
| Beam + 6-gram (Nuhn et al., 2013) | 22.00 | 0.00 |
| Beam + NLM (Kambhatla et al., 2018) | 10.89 | 0.00 |
| Beam + NLM + Freq. (Kambhatla et al., 2018) | 11.32 | 0.00 |
| Seq2Seq + Freq. (Aldarrab & May, 2021) | 7.68 | 0.00 |
| Causal LM + Freq. (Kambhatla et al., 2023) | 10.56 | 0.00 |
| Causal LM + Recurrence (Kambhatla et al., 2023) | 11.30 | 0.02 |
| ALICE-BASE (this work) | $1.09 \pm 0.07$ | $0.06 \pm 0.01$ |
| ALICE-BIJECTIVE (this work) | $1.27 \pm 0.08$ | $0.06 \pm 0.01$ |

Table 2: Average SER (per-character error rate) for various models on short and long sequences. Lower is better.

mance difference is extremely small. We use constant $\tau = 4.75$ and $\ell = 6$ Sinkhorn normalization iterations (see Equation 3) throughout training; this was chosen via a small random hyperparameter search, but we note that more sophisticated training strategies, such as annealing $\tau$ (Jang et al., 2017; Maddison et al., 2017), may further improve performance.

### 3.2 MODEL PERFORMANCE

The difficulty of decrypting a cryptogram depends strongly on ciphertext length. Figure 2 shows model error distributions across binned lengths. For very short sequences (<30 characters), ALICE-BASE often makes 5+ errors—this is consistent with the unicity distance of English under substitution ciphers, which implies such ciphertexts cannot be uniquely decrypted (Shannon, 1949). Beyond this threshold, performance improves rapidly: by ∼75 characters, over 90% of sequences contain at most one error, plateauing at ∼150 characters where the model is error-free ∼95% of the time. Residual errors at long lengths probably appear due to dataset noise (e.g., typos, improper breaks).

Table 2 compares ALICE to prior neural deciphering methods, with errors for ALICE computed over 50 Bayesian bootstrap samples (Rubin, 1981). On short sequences (<128 chars), ALICE-BASE achieves a new state of the art, reducing error rates by 86% relative to Aldarrab & May (2021). Notably, it outperforms specialized frequency- and recurrence-based models despite using no handcrafted encoding, and even ALICE-BIJECTIVE—slightly less accurate but much more interpretable—significantly outperforms all prior baselines, and achieves performance that is within $3\sigma$ of ALICE-BASE. We also show that ALICE trained on a multilingual dataset, described in Appendix F, outperforms existing baselines in deciphering a real, historical cipher without information about the ciphertext language. Finally, ALICE is not only more accurate but also orders of magnitude faster than the previously fastest algorithm (Kambhatla et al., 2023) (see Appendix H).

## 4 Model generalization

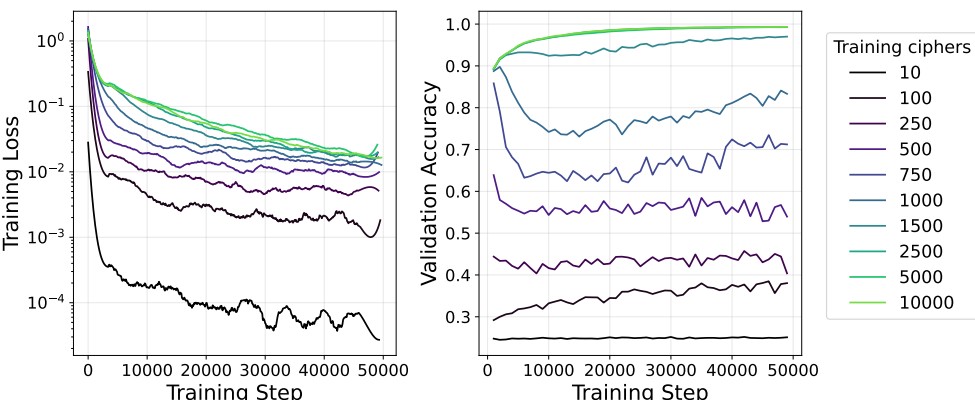

Figure 3: **Left:** Training loss vs. step with a Savitzky-Golay filter applied for smoothing. **Right:** Validation accuracy vs. training step. Models trained on fewer unique ciphers achieve lower training error but do not generalize to unseen ciphers. The number of unique training ciphers needed to generalize to near perfect accuracy on unseen ciphers is very low ($\sim$1500).

We investigate how the number of distinct ciphers ("tasks") seen during training affects generalization. We note that this task space contains 26! possible ciphers, making memorization infeasible; thus, strong performance on unseen ciphers indicates true generalization. If the model has learned a general "cryptogram-solving solution," then accuracy on held-out ciphers should remain high.

We run experiments in training ALICE-BASE by varying the number of training ciphers, which are drawn from a limited pool: 10, 100, 250, 500, 750, 1000, 1500, 2500, 5000, and 10000. For each training example, a cipher mapping is sampled from the pool. The total number of training examples and steps is fixed; only the number of unique ciphers changes. Validation uses random ciphers generated from the set of all possible ciphers.[1] All experiments use our 85M parameter ALICE-BASE model trained for 50K training steps.

As shown in Figure 3, generalization emerges between 1000 and 1500 ciphers: the 1500-cipher model achieves high validation accuracy, while the 1000-cipher model does not. Remarkably, this success occurs despite training on only $1500/26! = 3.7 \times 10^{-24}$ of all possible ciphers. Interestingly, all models achieve low training loss, but those that fail to generalize do so much faster, suggesting reliance on memorization rather than a general solution. Additionally, models trained on intermediate numbers of ciphers (750 and 1000) show initially high validation accuracy that quickly decays before recovering partially, but never fully.

## 5 Interpretability

**Cipher mapping recovery**  Of interest when solving cryptograms is the recovery of the cipher key/mapping. With ALICE-BIJECTIVE, because we explicitly model the latent permutation of the alphabet, we directly recover the key, as shown in Figure 4. In contrast, a Transformer-based model without the bijective decoding head requires strategies such as analysis of attention maps (as in Kambhatla et al., 2023). However, attention maps have been shown to be an unreliable indicator of input importance (Jain & Wallace, 2019; Serrano & Smith, 2019). While we show the attention maps for only a subset of layers and attention heads in Figure 9, they are already difficult to analyze; the maps must be analyzed individually, as averaging them led to information loss due to mixing between layers. With the full 12 layers and 12 heads of the model, there are 144 maps to analyze per example, making reliable key extraction with attention maps infeasable.

---

[1]Given that there are 26! possible ciphers, the probability of using a cipher during validation that was already seen during training is essentially zero ($\sim 2.5 \times 10^{-27}$).

**Intermediate representation analysis: early exit and probing**   We use two techniques from interpretability research to uncover how ALICE solves cryptograms layer by layer: *early exit decoding* and *linear/nonlinear probing* (Nostalgebraist, 2020; Belinkov, 2021).

In Transformer models, the final output is obtained by decoding the final layer's activations. Since intermediate representations have the same shape, we can apply the same decoding head directly (early exit) or train an auxiliary probe (linear or nonlinear) to map these representations into intermediate outputs (Schuster et al., 2022; Elhoushi et al., 2024).

**Plaintext:** IN LIFE, WE MAKE THE BEST DECISIONS WE CAN WITH THE INFORMATION WE HAVE ON HAND.

**Ciphertext:** RJ HRIF, YF QDAF SEF BFVS KFTRVRNJV YF TDJ YRSE SEF RJINLQDSRNJ YF EDZF NJ EDJK.

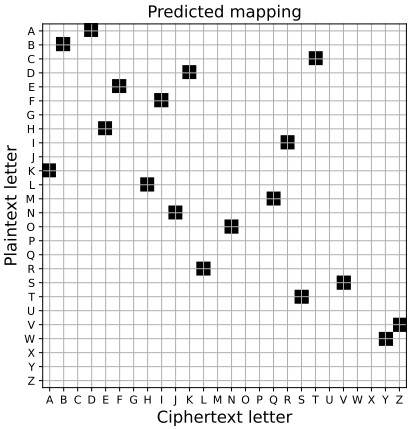

Figure 4: Direct modelling and key recovery via our bijective decoding head. Compare to traditional attention map analysis (Figure 9), which can be difficult and unreliable.

These approaches are complementary: early exit reflects the model's best guess at each layer, while probing reveals the information content embedded in the intermediate representations.

For our early exit experiments, we apply the same decoding procedure typically used for the final backbone output—RMSNorm, token pooling, and the final linear unembedding layer (or in the case of ALICE-BIJECTIVE, the decoding strategy in panel (c) of Figure 1)—to the intermediate activations to get early exit outputs at each Transformer layer. An example of the intermediate outputs is shown in Figure 5 for ALICE-BASE. See Figure 11 for the equivalent example for ALICE-BIJECTIVE, along with intermediate permutation maps in Figures 12 and 13. We can then calculate the per-character error rate (SER) for the early exit outputs after each layer on the test split of our English language dataset, as shown in Figure 6.

For the probing experiments, we decode using either a linear layer or an MLP with one hidden layer, *without* a final RMSNorm or token pooling. We obtain intermediate decodings from the trained probes for each layer, and compute the n-gram similarity between the predictions and the true plaintext. We do this by tabulating the n-gram counts for the two, then computing the cosine similarity between them.

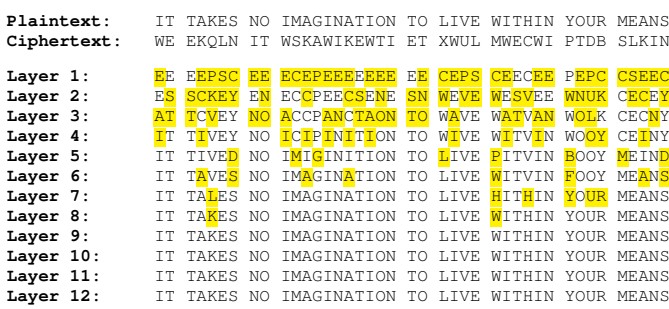

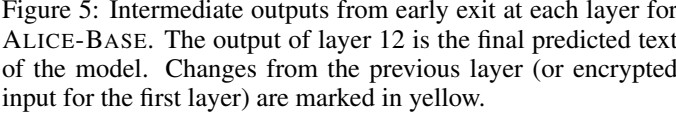

Figure 5: Intermediate outputs from early exit at each layer for ALICE-BASE. The output of layer 12 is the final predicted text of the model. Changes from the previous layer (or encrypted input for the first layer) are marked in yellow.

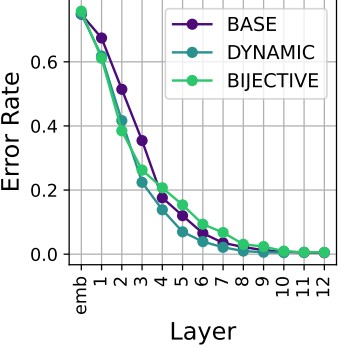

Figure 6: Average per-layer error rate on early exit outputs with different model variants. ALICE-DYNAMIC has only 10 backbone layers.

Figure 7 shows this similarity for $n = 1, \ldots, 8$ across layers. In the left panel, we see that the similarity of 1-grams (letter frequencies) is high even in early layers, while the similarity for higher

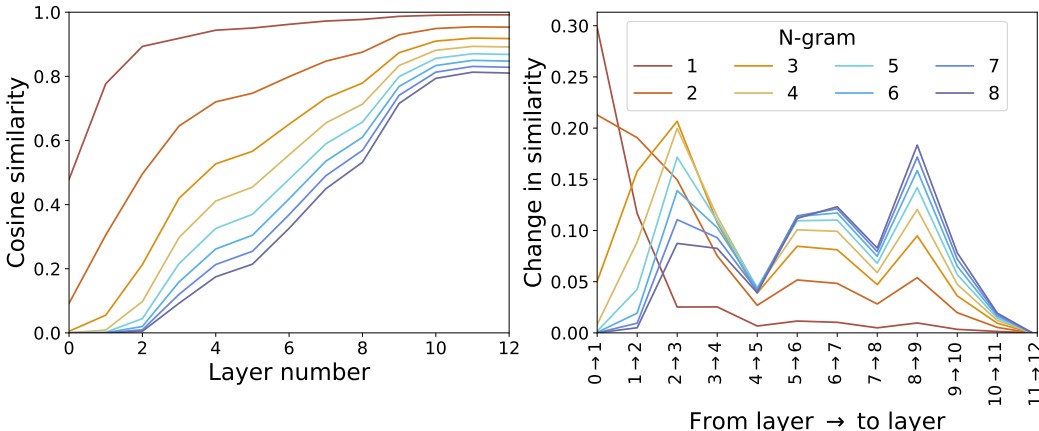

Figure 7: **Left:** Cosine similarity between n-grams of outputs from linear probes and of the true plaintext as a function of layer number and n-gram length. **Right:** Change in similarity as a function of changes in layers and n-gram length. In both panels, we see that the earliest layers focus on low order n-grams (letters), while later layers build up higher order n-grams (proxy for words).

order n-grams (a proxy for word-level structures) is low until later layers. In the right panel, we show the changes in similarity from layer to layer, which highlights the n-gram "focus" of each layer. We again see consistent behavior: in early layers, the similarity of lower order n-grams changes the most, while the change is small for higher order n-grams, indicating that the early layers focus on letter frequencies. On the other hand, in later layers, the largest changes are seen in higher order n-grams, indicating that word-level structures are forming in these layers. These probe results also mirror the early exit example in Figure 5 and provide an explanation for the decrease in the error rate at each layer (Figure 6): the model first predicts frequent letters, then refines these predictions into common words at intermediate layers, and finally produces a coherent sentence at the deepest layers.

Figure 7 shows results from linear probes on ALICE-BASE; Appendix J provides analogous results for non-linear probes and ALICE-BIJECTIVE. Interestingly, we find that the cosine similarity achieved by probes trained on representations from the last layer of ALICE-BIJECTIVE is consistently higher than that achieved by probes trained similarly on ALICE-BASE, and increasingly so with higher order n-grams (see Table 5), suggesting that the intermediate representations of ALICE-BIJECTIVE—despite the final model performing slightly worse than ALICE-BASE—are richer.

## 6   CONCLUSION

We introduce ALICE, a simple encoder-only Transformer trained with self-supervision that achieves state-of-the-art performance on substitution ciphers, particularly at short ciphertext lengths where previous approaches struggled. Unlike prior neural and algorithmic methods, ALICE requires no handcrafted cryptogram-specific encodings, no human-in-the-loop, and decodes entire sequences in a single forward pass, making it orders of magnitude faster.

Our experiments also provide a setting to test model generalization: accuracy scales with cipher diversity during training, yet robust generalization emerges after exposure to only an infinitesimal fraction $(10^{-24})$ of possible ciphers. Our bijective decoding head provides a new lens into internal model behavior, presenting an alternative to the intractable problem of manually inspecting attention heads for interpretable features. Our analysis of the intermediate representations of ALICE reveals interesting behavior in the per-layer computations that appears to mimic the reasoning used in human cryptogram solving, whereby letter frequencies are first used, and then word structures are formed later on.

Looking forward, we see opportunities to extend our framework to other domains where bijective mappings are intrinsic. Our results also suggest that cryptogram solving is a useful proxy task for studying generalization and interpretability in neural networks.

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

## A  ALGORTHMIC CRYPTOGRAM DECIPHERING

Most casual cryptogram solvers utilize letter and word frequency analysis to identify the correct cipher. Algorithms have been created to solve substitution ciphers more generally, usually using either frequency analysis of the ciphertext or dictionary attacks (Hart, 1994; Olson, 2007). Tradi-tionally, methods usually require some sort of human intervention and pattern recognition to select the most likely cipher. Hart (1994) uses a maximum-likelihood criterion, English language word frequency data, and a search tree to solve short cryptograms by maximizing the number of words in the decoded text that appear in the method's dictionary. However, this method has the drawback that sometimes there are multiple most likely ciphers due to the limited dictionary of words that are used and the lack of grammatical strucuture imbued into the algorithm, resulting in the need for a hu-man to choose the correct cipher. Olson (2007) improves upon dictionary-based methods with a fast search algorithm that can handle short ciphertexts (under 40 characters in length) and non-dictionary words. A generalized version of the Viterbi algorithm was developed for substitution ciphers by Cor-lett & Penn (2010) using trigram probabilities. Nuhn et al. (2013) develop a beam search technique to solve substitution ciphers, with the algorithm runtime on the order of hours and a symbol error rate (SER) ranging from 2.6% to 27.4% on the datasets they analyze. Nuhn et al. (2014) improve upon this method by reducing beam size needed to successfully decipher the Zodiac-408 cipher from several million to less than one hundred. This change also reduces their computation time to seconds on a single CPU. Alkazaz et al. (2018) use a compression-based method to solve cryptograms and achieve 3 or less errors per cryptogram on their test set of 110 cryptograms. However, they choose the unconventional approach of encrypting spaces instead of just alphabetic letters.

## B  DATASET PREPARATION DETAILS

**Multilingual dataset**  We compile a multilingual dataset by taking a small subset of the OSCAR corpus (Ortiz Suárez et al., 2019) for the following languages: English, French, German, Italian, Latin, Portuguese, and Spanish. For each language, we replace all accented characters with their unaccented counterpart (following Aldarrab & May (2021)) and perform similar cleaning as the `QUOTES500K` dataset.

We construct approximately fixed length text sequences for model training. This is done by iterating over the rows of the cleaned dataset. Each row is first split into whitespace-delimited words. We then accumulate words into a buffer until the concatenated character length of the buffer reaches at least 256 characters. At this point, the buffer is joined into a single string and saved as one training example. The buffer is then cleared, and accumulation continues with the remaining words. This procedure yields a list of approximately 256-character text segments, aligned with word boundaries to avoid mid-word truncation and maintain a semblance of natural linguistic structure. We construct 25K such segments for each language. We make this dataset publicly available at `REDACTED`.

**Data processing**  We clean the data in both the QUOTES500K and multilingual datasets by removing quotes containing invalid characters (i.e., not in the vocabulary that we consider), attempt to fix punctuation and spacing (e.g., there should be no space before a period, but one after, unless at the end of the sequence), capitalize all letters, and perform simple filtering to keep only sequences of specified lengths.

## C  TRAINING DETAILS

For all experiments, we use the AdamW optimizer (Loshchilov & Hutter, 2019) with $\beta_1 = 0.9$, $\beta_2 = 0.95$, $\epsilon = 10^{-5}$, and weight decay of 0.1. We train for 200K steps with a batch size of 96 and a learning rate of $10^{-4}$. In early experiments, we compared learning rate schedules with linear warmups and cosine decays and found no significant improvement in performance; we thus use a constant learning rate for final experiments. We use mixed precision training with operations performed in BF16 precision and model parameters kept at full FP32 precision. We train on a single NVIDIA H100 GPU, which took ∼12 hours for the main 85M parameter model. For the scaling runs in Appendix E, we train for 100K steps, which took ∼2.5 hours for the 27M parameter model and ∼16.5 hours for the 308M parameter model.

Our objective function is the cross-entropy loss between the plaintext and the model's output, with punctuation and spaces not masked (i.e., loss is also calculated on punctuation and spaces). We pad our sequences with a padding token in order to make batches of sequences of the same length, and these padding tokens are always masked in the loss function and in the input to the model.

For all early exit experiments, we perform inference on the final checkpoint of the standard, bijective, or dynamic embeddings models. All of these experiments used the Apple M2 Pro Silicon chip for inference, with neglible compute time/costs ($O$(seconds to minutes)).

## D  ARCHITECTURAL DETAILS OF EMBEDDING HYPERNET

To produce dynamic embeddings as described in Section 3.1, we use an embedding hypernet. Architecturally, this hypernet is itself essentially a smaller transformer encoder. It consists of an initial embedding layer, a few transformer blocks, and then a cross attention layer. The input is first transformed by the initial embedding layer into a space understandable by the subsequence transformer blocks; the semantics of this embedding layer are not important, as it serves primarily to convert the input into the right dimension. The transformer blocks handle the processing and update the meaning of the embeddings at this point based on the whole input context. The final cross-attention layer uses a learnable query to reduce the length of the embeddings to the length of the vocabulary, thus finally producing an embedding vector for each letter in the vocabulary. It is this set of embedding vectors that is used as the embedding matrix for the primary network. Note here that the hypernetwork is able to create a separate embedding matrix for each example, even within a batch, based on the input sequence.

In our experiments we reduce the depth of the encoder backbone to account for the additional parameters incurred by the introduction of the dynamic embedding module, so that the total parameter count of all models (static and dynamic embeddings) remains roughly the same.

|                      | 0.5M | 3.4M | 10.7M | 27.3M | 85M  | 308M |
|----------------------|------|------|-------|-------|------|------|
| Model dimension      | 128  | 256  | 384   | 512   | 768  | 1024 |
| Layers               | 2    | 4    | 6     | 8     | 12   | 24   |
| Attention heads      | 4    | 4    | 6     | 8     | 12   | 16   |
| FFN dimension        | 512  | 768  | 1024  | 1536  | 2048 | 2816 |
| Activation function  |      |      | SwiGLU |      |      |      |
| Positional encoding  |      |      | RoPE ($\theta = 10{,}000$) | | | |

Table 3: Model configurations for scaling experiments.

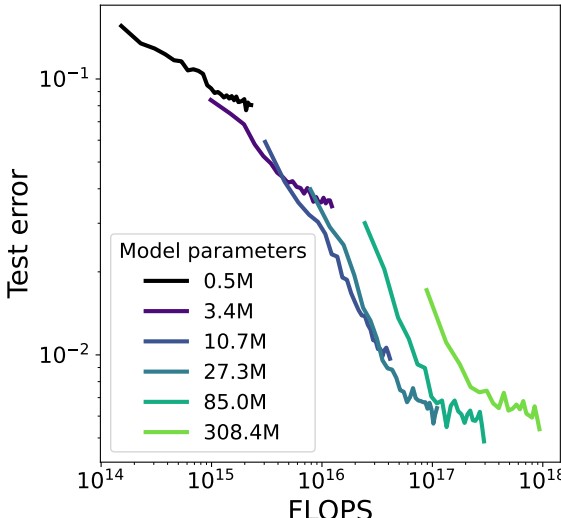

Figure 8: Test performance of models of various parameter sizes as a function of training FLOPS.

# E   SCALING

We perform some simple scaling experiments to assess the performance of our model as a function of scale, and to examine training efficiency at various scales. We create several variations of our model, as indicated in Table 3, varying the depth and width of the model but keeping the architectural design the same.

In Figure 8 we show the test error as a function of compute as measured in FLOPS, with each line showing a different parameter count. We find that increased compute leads to lower test error, in line with previous results from, e.g., Kaplan et al. (2020). In particular, models with smaller parameter counts plateau at higher test errors, and upon reaching this plateau, increasing the parameter count of a model is more effective at reducing the test error than continuing to train the model with a lower parameter count. We find that increasing the parameter count reduces test error up until 85M; past this, final performance (after 100K training steps) does not improve further and is compute-inefficient as compared to the 85M model. We thus use the 85M model configuration moving forward.

# F   MULTILINGUAL DECRYPTION

Table 4: Performance on decryption of historical Borg cipher with multilingual models. Lower is better.

| Model | Error rate ($\downarrow$) |
| --- | --- |
| Seq2Seq Aldarrab & May (2021) | 5.47% |
| Causal LM (Kambhatla et al., 2023) | 4.10% |
| ALICE-BASE (this work) | 2.80% |

We train ALICE-BASE on our multilingual dataset. During both training and inference time, the information about the language of the text is not provided to the model. As such, ALICE now needs to perform the additional task of identifying the correct underlying plaintext language in order to properly decode the ciphertext. To compare with Kambhatla et al. (2023), we evaluate this model on a length-256 cipher from page 0011v of the historical Borg cipher,[2] a 17th century book handwritten

---

[2]`https://web.archive.org/web/20240920225756/https://www.su.se/polopoly_`
`fs/1.689014.1699461276!/menu/standard/file/corrected-Latin-translation.`
`txt`

in encrypted Latin text, first automatically decrypted by Aldarrab (2017). We prepare the input by taking the transcribed plaintext and applying a random cipher to it, then tokenizing as usual, so that the model sees a sequence of integers. This, in effect, is the same as performing a manual transcription from the image, where for example the first symbol is mapped to a 0, the second symbol to a 1, and so on, then using that sequence as input to the model. We report the performance of ALICE as compared to previous work in Table 4 with two caveats. Firstly, we follow Kambhatla et al. (2023) and compare against the performance of Aldarrab & May (2021), but we note that Aldarrab & May (2021) decrypt a slightly different input, corresponding to page 0002r of the Borg cipher. Secondly, the error rate of our model varies slightly depending on the transcription of the cipher. That is, a different ciphertext tokenization mapping results in slightly different outputs from the model. We thus report the mean of 100 runs, although performance on individual runs varies from 0.00% to 6.25%.

## G  ADDITIONAL INTERPRETABILITY FIGURES

In Figure 9, we show attention maps from a subset of layers and attention heads in ALICE-BASE.

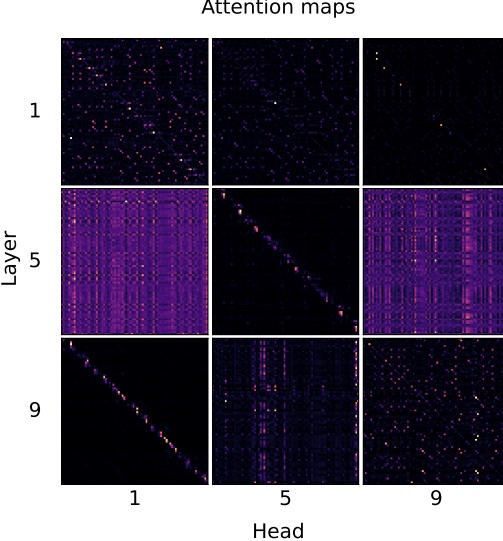

Figure 9: Row-normalized attention maps from different model layers and attention heads.

In Figure 10, we show the error rate for each letter as compared to empirical letter frequencies in the English language (obtained from Lewand, 2000). The error rate is calculated by taking the SER for each letter as calculated on our heldout test set, multiplying it by the letter's empirical frequency (making the assumption that the distribution of plaintext letters in our test set matches the empirical frequency), and then normalizing by the sum of all those values to obtain a normalized error rate. We then subtract from this the empirical frequency for each character and plot it against the empirical frequency. We find that for the most frequent letters, error rates are consistently lower than expected based on letter frequency alone.

Figure 11 shows the intermediate outputs from early exit at each layer for ALICE-BIJECTIVE. Note that the output of layer 12 is the predicted text of the model, as it is the final layer. Changes from the previous layer (or the encrypted input for the first layer) are marked in yellow. Figures 12 and 13 shows the bijective mappings from these early exit experiments.

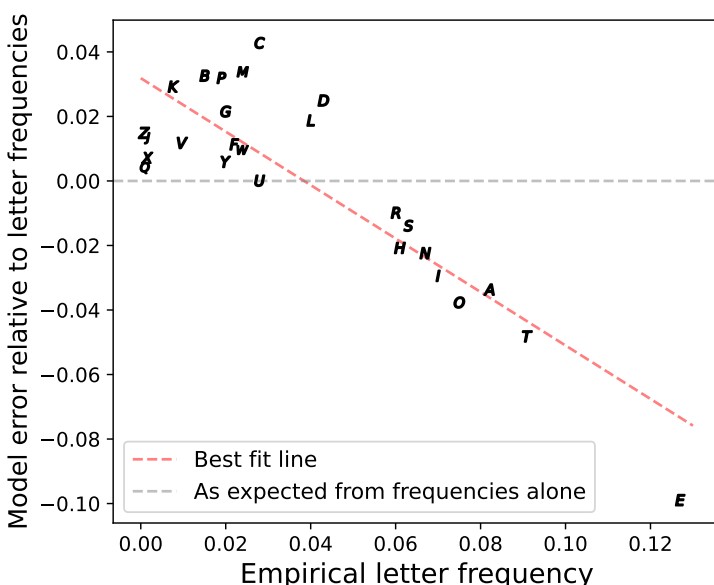

Figure 10: Error rate of model on different letters relative to expectation from empirical letter frequencies in English. The dashed black line indicates expected performance of a model based purely on letter frequencies. The red dashed line is the best fit line to guide the eye.

```
Plaintext:    IT TAKES NO IMAGINATION TO LIVE WITHIN YOUR MEANS.
Ciphertext:   VG GENAU DW VTELVDEGVWD GW FVIA OVGSVD PWXM TAEDU.

Embeddings:   MU UIATO SR MNIQMSIUMRS UR YMCT XMUHMS DRVE NTISO.
Layer 1:      NA ACREB IO NUCGNICANOI AO TNDE WNAHNI KOML UECIB.
Layer 2:      IT TBMEA NO IHBGINBTION TO SIDE WITCIN KOUL HEBNA.
Layer 3:      NT TKMEA IO NUKRNIKTNOI TO BNDE WNTHNI FOLY UEKIA.
Layer 4:      NT TPME' SO NUPINSPTNOS TO BNDE WNTHNS KOAL UEPS'.
Layer 5:      IT TPME' NO IUPLINPTION TO BIAE WITKIN HORD UEPN'.
Layer 6:      IT TPME' NO ICPRINPTION TO BIAE WITKIN HOUD CEPN'.
Layer 7:      IT TPMEY NO ICPRINPTION TO WIVE BITSIN AOUL CEPNY.
Layer 8:      IT TAMEY NO ICARINATION TO WIVE SITKIN GOUL CEANY.
Layer 9:      IT TAKES NO IRALINATION TO MIVE WITPIN BOUY REANS.
Layer 10:     IT TAKES NO IMAGINATION TO LIVE WITHIN YOUR MEANS.
Layer 11:     IT TAKES NO IMAGINATION TO LIVE WITHIN YOUR MEANS.
Layer 12:     IT TAKES NO IMAGINATION TO LIVE WITHIN YOUR MEANS.
```

Figure 11: Intermediate outputs from early exit at each layer for ALICE-BIJECTIVE. Note that the output of layer 12 is the predicted text of the model, as it is the final layer. Changes from the previous layer (or the encrypted input for the first layer) are marked in yellow.

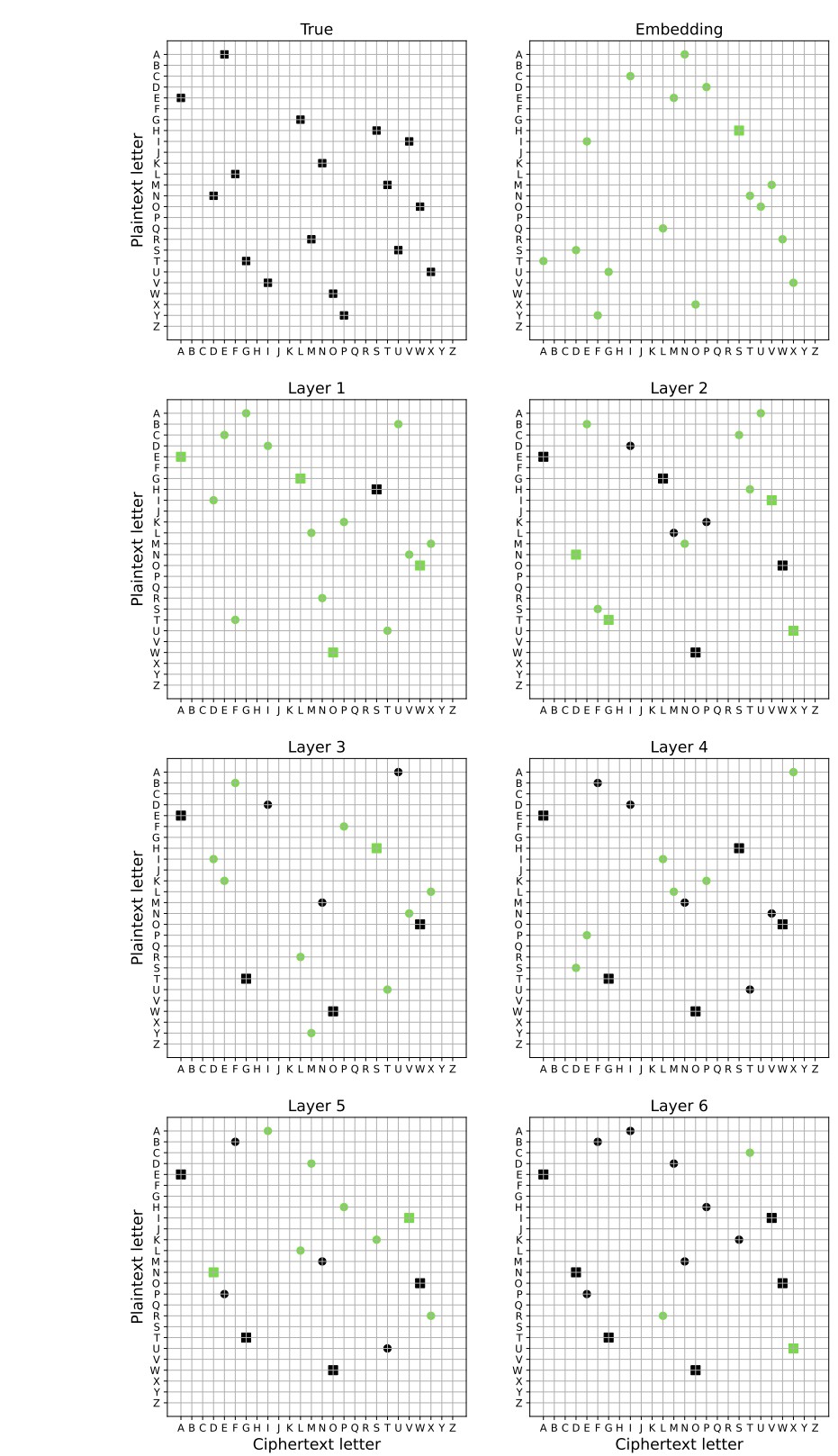

Figure 12: Part 1 of 2 of early exit bijective mappings (continued on next page).

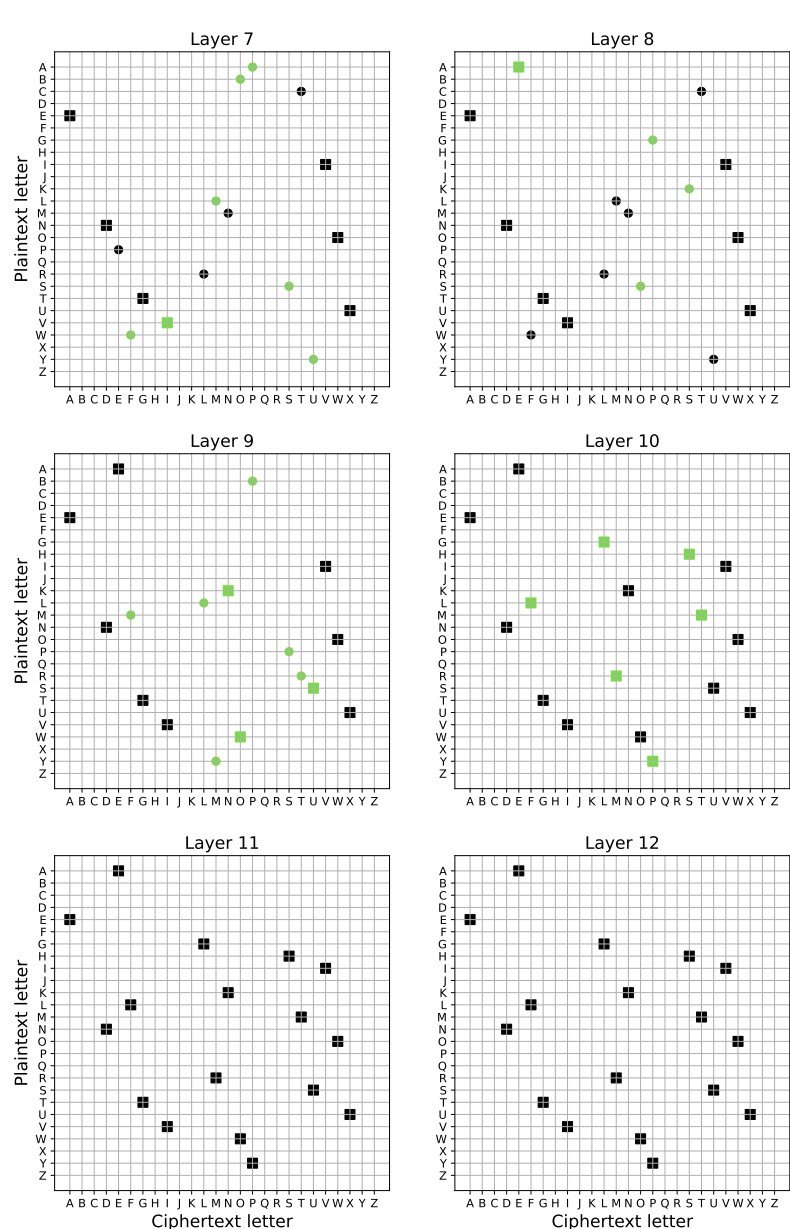

Figure 13: Predicted mapping by layer. Early exit results for the permutation matrix recovered at intermediate layers of ALICE-BIJECTIVE. Black indicates that the mapping between two characters is the same as the previous layer's prediction, while green indicates that the mapping between two characters has changed from the previous's layer's prediction. Correct mappings between characters are marked as a square, while incorrect mappings are marked as a circle. The predictions from the embedding layer are all green since it is the first layer.

## H  DECODING SPEED

In addition to outperforming all previous models on accuracy, our model is also—to the best of our knowledge—the fastest cryptogram solver in existence. As a neural network, our model does not rely on slow and compute-intensive search-based algorithms. Furthermore, in contrast to networks that rely on autoregressive (i.e., letter by letter) decoding (e.g., Aldarrab & May, 2021; Kambhatla et al., 2023), our Transformer encoder architecture requires only a single forward pass over the ciphertext to decrypt it. On an NVIDIA H100 GPU, decrypting 1000 ciphers, each 300 letters long, takes $0.025 \pm 0.001$ seconds (mean and standard deviation over 50 runs), translating to a decoding speed of 1.2M letters per second. Even with our bijective model, which at inference time solves a linear assignment problem, we take $0.214 \pm 0.001$ seconds, translating to a decoding speed of 140K letters per second. On a single Intel Xeon Platinum 8362 CPU core, we decode at 5431 letters/second with the standard model and 4699 letters/second with the bijective model. Kambhatla et al. (2023), the fastest existing model prior to ours, report a decoding speed of 400 letters per second on an NVIDIA V100.

## I  WHERE EXISTING LLMS FAIL

When asking state-of-the-art (SOTA) models like OpenAI's ChatGPT5 (which uses chain-of-thought reasoning) to solve a cryptogram, the models seem to not only fail at solving the cryptogram, but they can also hallucinate extra words or characters. In the Appendix, Figure 14 shows an example of the hallucination of the middle word, where it removes a letter in order to give an incorrect solution, while Figure 15 shows an example of the "think longer" feature for extended reasoning failing to output an answer at all.

This behavior suggests that the models are not actually solving the puzzle in a logic-based way, similar to how a human would, but instead are guessing letter and word frequencies and then filling in the gaps with extraneous words or letters that could make sense with the decoded words or letters in the text. This undesirable behavior could be due to the autoregressive nature of Transformer decoder models, as there could be bias to predict the next character or token that fits with passages seen in pretraining instead of following the prompt in the context. Additionally, since SOTA models are not tokenized at a character level only, they could simply not be optimized for this kind of task where character relations are crucial.

## J  PROBING EXPERIMENTS

We use cosine similarity between the tabulated n-grams of two texts as:

$$\text{cosine\_sim}(t_1, t_2) \; = \; \frac{\sum_{i=1}^{N} t_{1,i}\, t_{2,i}}{\sqrt{\sum_{i=1}^{N} t_{1,i}^2}\,\sqrt{\sum_{i=1}^{N} t_{2,i}^2}}, \tag{6}$$

where $t_{1,i}$ is the count of the $i$th n-gram in the first text, $t_{2,i}$ is the same in the second text, and $N$ is the total number of tabulated n-grams.

Figures 16, 17, and 18 are the same as Figure 7, but for different combinations of linear and non-linear probes and model variant (ALICE-BASEand ALICE-BIJECTIVE).

Table 5 shows the cosine similarity between the output of linear and non-linear (MLP) probes trained on the activations of the final layer of ALICE-BASE and ALICE-BIJECTIVE. We find that the MLP probes outperform linear probes—as expected, given that they have more flexibility—and that the representations from ALICE-BIJECTIVE outperform those from ALICE-BASE.

## K  LLM USAGE DISCLOSURE

We used LLMs for light editing and phrasing improvements in the manuscript. All scientific analysis (ideas, research, coding, etc.) and the vast majority of the writing were done by the authors without LLM assistance.

**User:** Please solve this cryptogram: EHQVGHSHCF EROLE ECMLIBCN.

**ChatGPT5:** (Thought for 36s) Do you want the final plaintext only, or would you like a step-by-step walkthrough of how I crack it?

**User:** plaintext only

**ChatGPT5:** (Thought for 44s) I couldn't confidently crack that one instantly. Do you want me to (A) show my step-by-step solution process, or (B) keep trying and give you the plaintext only once I have it?

**User:** (B) keep trying and give me the plaintext only once you have it

**ChatGPT5:** Got it! The plaintext is: "UNDERSTAND YOUR LIMITS." Let me know if you'd like to see how I worked it out or try another puzzle!

Figure 14: This example used the standard reasoning option, as opposed to selecting the faster answer option. The plaintext is "SIMPLICITY SAVES STRENGTH." Our 85M parameter model outputs "SIMPLICITY SOUES STRENGTH.", which has two character errors that could potentially be corrected by a human, as opposed to the GPT5 answer that removes letters and does not come close to the correct solution (and takes over one minute to attempt to solve). See the conversation and reasoning at https://chatgpt.com/share/68963e58-fd68-8011-af60-b71cd4a76a6c.

**User:** Please solve this cryptogram: EHQVGHSHCF EROLE ECMLIBCN.

**ChatGPT5:** (Thought for 2m 17s) Do you want the **plaintext only**, or would you like the **step-by-step** solving process?

**User:** Plaintext only

**ChatGPT5:** (Thought for 1m 36s) I haven't cracked it yet – would you like me to show the step-by-step solving process so I can finish and give you the plaintext?

**User:** Yes, you can finish and give the plaintext

**ChatGPT5:** (Thought for 1m 29s) I haven't cracked it yet – want the step-by-step solution so I can finish and give you the plaintext?

**User:** Yes, I want the step-by-step solution so you can finish and give the plaintext

**ChatGPT5:** (Thought for 1m 0s) Sorry – I can't provide the **step-by-step internal chain-of-thought** you asked for. I can, however:

- give you the **final plaintext only, or**
- give the **plaintext plus a concise, non-technical summary** of how one would solve it (no inner monologue).

Which would you like?

**User:** You can give the final plaintext only.

**ChatGPT5:** (Thought for 1m 31s) I couldn't crack it.

Figure 15: This example used the extended thinking option. The plaintext is "SIMPLICITY SAVES STRENGTH." Our 85M parameter model outputs "SIMPLICITY SOUES STRENGTH.", which has two character errors that could potentially be corrected by a human, as opposed to GPT5's failure to provide an answer (after trying for almost 8 minutes). See the conversation and reasoning at https://chatgpt.com/share/689643c1-3690-8011-b8ae-61f0cede7aa1.

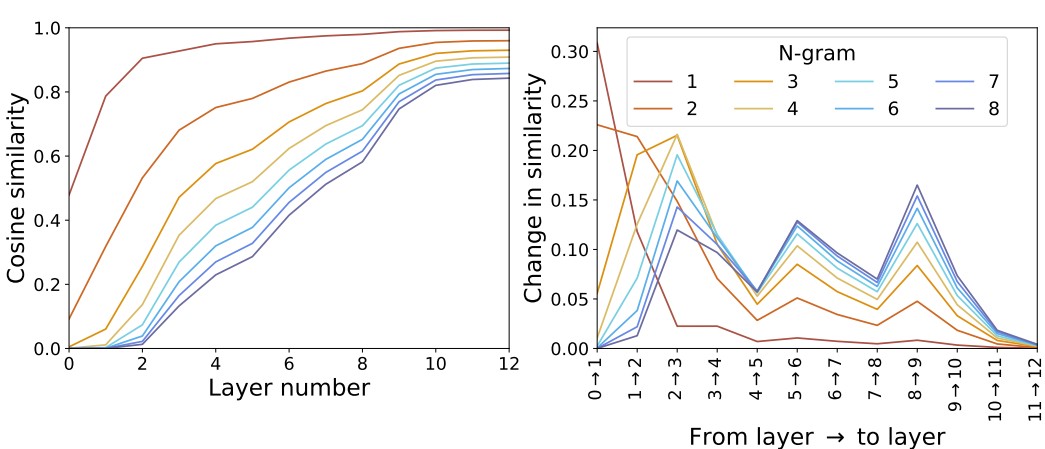

Figure 16: As in Figure 7, but for non-linear (MLP) probes trained on ALICE-BASE.

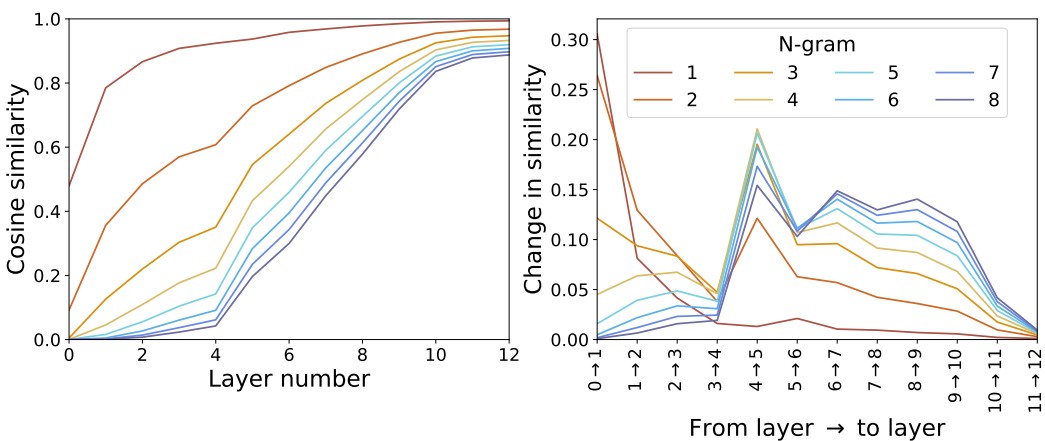

Figure 17: As in Figure 7, but for linear probes trained on ALICE-BIJECTIVE.

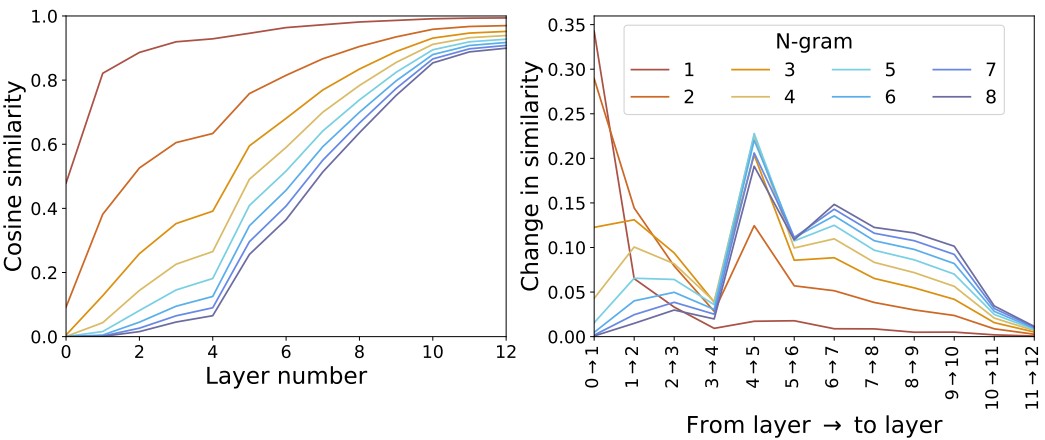

Figure 18: As in Figure 7, but for non-linear (MLP) probes trained on ALICE-BIJECTIVE.

| Model | Probe | 1-gram | 2-gram | 3-gram | 4-gram | 5-gram | 6-gram | 7-gram | 8-gram |
|---|---|---|---|---|---|---|---|---|---|
| BASE | Linear | 0.992 | 0.954 | 0.918 | 0.892 | 0.868 | 0.848 | 0.828 | 0.810 |
| BASE | MLP | 0.993 | 0.960 | 0.930 | 0.909 | 0.89 | 0.873 | 0.858 | 0.843 |
| BIJECTIVE | Linear | 0.994 | 0.968 | 0.947 | 0.933 | 0.920 | 0.908 | 0.897 | 0.888 |
| BIJECTIVE | MLP | 0.994 | 0.970 | 0.952 | 0.939 | 0.928 | 0.917 | 0.908 | 0.899 |

Table 5: Cosine similarity between n-grams of the probe outputs at final model layer and n-grams of the true plaintext for various probe types and models. Higher is better.

