# OpenReview forum: "Alice: An Interpretable Neural Architecture for Generalization in Substitution Ciphers"
_ICLR.cc/2026/Conference — ICLR 2026 Conference Withdrawn Submission_

### Official Review · Reviewer_Vaqv · 2025-10-24

**Soundness:** 2
**Presentation:** 3
**Contribution:** 2
**Rating:** 4
**Confidence:** 4

**Summary:**

This paper investigates how Transformer models can learn to decode monoalphabetic substitution ciphers. The authors propose a pooling-based decoder, and a bijective decoding head designed to enforce one-to-one token mappings. The proposed method achieves state-of-the-art performance on short cipher sequences and contributes an intuitive and efficient architectural component for modeling 1:1 substitutions.

**Strengths:**

1. The pooling method is intuitive and effective, and the paper proposes a bijective decoding head to enhance interpretability.
2. The approach achieves state-of-the-art performance on short 1:1 ciphertext sequences.

**Weaknesses:**

The task studied is relatively simple, and the interpretability of how the model solves it remains incomplete.
1. The 1:1 monoalphabetic substitution cipher is a simple problem, and it is simpler than the homophonic ciphers already studied in Kambhatla et al., 2023, which allow multiple ciphertext symbols per plaintext character and flatten frequency distributions.
2. The main innovation of this paper is the bijective decoding head, but it does not improve model performance. The claimed interpretability benefit also seems limited: even without a bijective head, the cipher map can still be easily visualized. The only difference is that the resulting mapping is no longer strictly monoalphabetic.

**Questions:**

I am curious what the average Symbol Error Rate (SER) curve looks like as a function of ciphertext length, like figure 1 in Kambhatla et al., 2018. It would help clarify how quickly the model begins to recover the cipher mapping, especially on very short sequences.

---

> ### Author Response · Authors · 2025-11-28
>
> We thank the reviewer for taking the time to carefully assess our paper.
>
> ## Weaknesses
>
> 1. We agree that the 1:1 monoalphabetic substitution cipher is a simpler problem than many/most other ciphers; we study this task precisely for its simplicity, which makes it a good testbed for conducting interpretability experiments on our model. Interesting future work could be to extend our analyses to more complex ciphers.
> 2. A cipher map can indeed be recovered for a model without the bijective head, but the difference highlighted by the reviewer—"the resulting mapping is no longer strictly monoalphabetic"—is an important one, because that violates the key assumption made in the task. A model should not be making a mistake of this type in the first place (for example, a human solving puzzles of this kind will never make a mistake like this). Explicitly enforcing bijectivity ensures that you will always get a structurally valid solution (and a structurally valid mapping), even if the SER (symbol error rate) is very slightly lower.
>
> ## Questions
>
> We refer the reviewer to Figure 2, where we show a detailed breakdown of the number of errors made by the model as a function of ciphertext length.

---

### Official Review · Reviewer_hvsH · 2025-10-24

**Soundness:** 2
**Presentation:** 3
**Contribution:** 1
**Rating:** 2
**Confidence:** 4

**Summary:**

This paper employs encoder-only Transformer models to solve substitution ciphers and demonstrates that their performance on short-length ciphers surpasses that of existing algorithms. In parallel, the authors perform an interpretability analysis, revealing that the model has successfully learned the underlying substitution rule. Finally, they apply a linear probe to analyze how the model encodes the learned functionalities.

**Strengths:**

1. The paper is written clearly with a proper description of the experiments.
2. Source code was provided.

**Weaknesses:**

1. The baseline comparison is misleading. For instance, the cited work by Kambhatla et al. (2023) uses a significantly larger dictionary size than the authors’ setup, which invalidates the claimed SOTA performance.
2. The authors repeatedly emphasize that the search space in their setting is $26!$, but this is not how simple substitution ciphers work. I would encourage them to check standard references such as https://cacr.uwaterloo.ca/hac/about/chap7.pdf and the concept of unicity distance. From theoretical considerations, about 28 characters are needed to brute-force such a cipher for English. Even from a data perspective, the dataset used in this work already contains more than enough samples to fully recover the character frequency distribution of the plaintext.
3. From an AI perspective, the authors do not demonstrate the scalability of their algorithm. Particularly, for larger dictionary sizes, which are both more challenging and more practically relevant. This issue is closely related to my first point.
4. The interpretability study is limited. For example, line 368 states: “because we explicitly model the latent permutation of the alphabet, we directly recover the key, as shown in Figure 4.” This statement is essentially tautological to me: “because we explicitly model the latent permutation, we can extract the permutation.” Moreover, while the linear probe and n-gram analyses show that the model progressively solves the task, the authors do not provide any mechanistic explanation for how this happens, relying instead on vague verbal reasoning.

**Questions:**

On top of the weaknesses mentioned above, I have a few more detailed questions for the authors:

1. Why do the authors average features from the same character at different positions? Wouldn’t simply removing positional embeddings achieve the same goal, as this naturally enforces positional invariance? An encoder-only architecture does not introduce imbalance in-context like decoder-only models.
2. Figure 9 might not be as difficult to interpret as suggested. Layer 1, head 1 already appears to capture similarities between identical characters and some substitution patterns. I also recommend computing cosine similarities or correlations of internal features across different characters, which would provide a straightforward observable for how the model learns the deciphering algorithm layer by layer.
3. Do we really need 12 layers? A smaller model should be able to solve this task and might be more interpretable and faster to run.
4. It would be very interesting if the authors could increase the dictionary size or vary the language distribution and evaluate whether their conclusions continue to hold under these more challenging and realistic conditions.

---

> ### Author Response · Authors · 2025-11-28
> **part 1 (weaknesses)**
>
> We thank the reviewer for their insights and appreciate the detailed questions. We hope to address these questions below:
>
> ## Weaknesses
>
> 1. While Kambhatla et al. (2023) analyzes homophonic ciphers with a larger dictionary size, the results we are comparing to are from Section 6 of their paper, where they study simple substitution ciphers with the same setup as ours. In particular their dictionary size is no larger than ours. We refer the reviewer to Table 5 of Kambhatla et al. (2023).
>
> 2. We would like to ask the reviewer to please explain why they think that the search space for simple substitution ciphers is not 26!. We have looked through the reference that was provided by the reviewer, and from Example 7.73 on Page 247 we quote:  "The number of keys for a mono-alphabetic substitution cipher over alphabet A is |K| = s!, where s = |A|. For example, s = 26 (Roman alphabet) yields $26! \approx 4 \times 10^{26}$ keys." With regards to the unicity distance, we agree with the reviewer, and point to Section 3.2 in our manuscript, where we have explained this and have cited Shannon (1949), which is the original paper that coins the term unicity distance. We agree with the reviewer that the dataset provides strong constraints on the letter frequency distribution, and that by analyzing language statistics, we can gain some information about how to solve the ciphers. This is the basis of our n-gram analysis in our interpretability section. We note, however, that simply knowing only the letter frequency distribution may not be enough to solve the cryptograms in many cases.
>
> 3. In regards to scaling up the size of the dictionary, we agree that this would be an interesting direction to probe. While we agree that more challenging ciphers with a larger dictionary size may be intriguing to analyze, here we study 1:1 monoalphabetic substitution ciphers precisely because it is a simpler problem than many/most other ciphers, which makes it a good testbed for conducting interpretability experiments on our model. It would certainly be interesting for future work to extend our analyses to more complex ciphers. For the scope of this work, we instead decided to investigate model performance when scaling up the number of languages learned at one time, requiring the model to learn multilingual language distributions. However, we follow the precedent of previous work by Aldarrab and May (2021) and strip special characters so the multilingual data conforms to the English alphabet. Other extensions could include analyzing languages that have larger alphabets or include these special characters as additional letters in the alphabet.
>
> 4. The reviewer highlights the simplicity of recovering a permutation with our bijective model; the permutation is explicitly modelled, and so there is no additional work to do in order to recover it. This is in contrast to other models, where additional work must be done in order to recover the implicitly modelled permutation. If the reviewer means that we can extract the permutation anyways by looking at the inputs and predicted outputs, we point to our response to Weakness #2 from reviewer Vaqv. With regards to interpretability, we argue that the n-gram analysis shows not only that the model progressively solves the task, but also how it solves the task. In particular, we can break down the solving process into a series of steps, each corresponding to a given layer in the Transformer. Figure 7 (in particular, the right panel) shows quantitatively, and step by step, how the task is solved, namely by building up progressively higher order n-grams at each layer.

---

> ### Author Response · Authors · 2025-11-28
> **part 2 (questions)**
>
> ## Questions
>
> 1. As described in the text, we average features from the same character at different positions in order to ensure consistency between the outputs for a given input letter. As in, it ensures that, for example, two A's in the ciphertext will always map to B in the predicted output, instead of one A mapping to B and one to C in the same cipher. This is not the same as positional invariance. Removing positional embeddings would not make sense, because it would be as if we fed a scrambled list of letters to the model without any semantic structure. When trying to construct a sentence, it matters what order the letters are in, and we want to preserve that order. Even if the individual letters have been changed, the underlying sentence still has structure.
>
> 2. While we agree that Figure 9 shows some repeated patterns in the attention heads, it is difficult to deduce the algorithm or mechanism by which the Transformer is solving the cipher. We are also still unsure of how to interpret these attention maps automatically, which early exit and our n-gram analysis allows for. We appreciate the suggestion to compute cosine similarities or correlations between internal features of the model, but this still does not allow for an interpretation of what these features mean or represent. For instance, a high cosine similarity between neurons, attention heads, or MLPs calculated across different characters doesn’t translate into an algorithm or circuit that produces the model’s behavior, just that certain components activate strongly in response to an input. The combination of these features into a circuit would require extensive additional experiments.
>
> 3. We refer the reviewer to Figure 8 in Appendix E, where we show how the number of layers and the size of the model affects performance. We find that while smaller models are able to solve the task, we achieve the best performance with a 12-layer 85M parameter model. The inference cost of the 85M parameter model is so small that we find little benefit to reducing model size (at the cost of performance) in order to save time. As mentioned in Appendix H, on modern hardware we can decrypt 1.2M tokens/second with the 85M parameter model, and we see little need to go faster.
>
> 4. We refer the reviewer to Appendix F (also referenced in Section 3.2 (line 321)), where we analyze a model trained on a multilingual dataset composed of 7 different languages. This multilingual model was trained simultaneously on these 7 languages, without knowing which language the ciphertext came from. Further details about the preparation of the multilingual data can be found in Appendix B.

---

### Official Review · Reviewer_qtE9 · 2025-10-26

**Soundness:** 2
**Presentation:** 4
**Contribution:** 1
**Rating:** 2
**Confidence:** 2

**Summary:**

The paper is concerned with the cryptogram task: given a bijection between alphabets, applying the bijection transforms a text to a cipher text. One needs to recover the original text from the cipher text.
The paper introduces an encoder-only transformer architecture that, given a training dataset consisting of a pair of (text, ciphertext), it solves the cryptogram task in a data efficient way that is claimed to be the state-of-the-art. In addition, it further explored the interpretability angle and presented some findings in interpreting the networks' prediction mechanism.

**Strengths:**

Clear presentation and thoughtful ablations. The paper itself is a good one, and is fairly complete in terms of a proper research paper which deserves to be published somewhere. It is just debatable how good the problem and solutions are, and how much new insights does it tell us so that it can be presented in this venue.

**Weaknesses:**

Lack of novelty. There isn't much insight that is new. Also I'm not sure if I miss anything, but the difficulty and significance of the work appears to be significantly exaggerated in the abstract and some of the main text.
Also, the task could have been generalized and pitched a lot more toward interpretability research. But to be fair, the paper did make some interesting effort using linear probing.

**Questions:**

Is Figure 2 the only justification for the state-of-the-art status? Are you trying to compare with "any" other methods without constraints, or has there been certain limit on the amount of compute spent at either training time and/or serving time? As to the SoTA status, I do not mean that every paper needs to chase SOTA, but you did make the this claim in the abstract: "... that sets a new state-of-the-art for both accuracy and speed on this decryption problem". And typically in this scenario, constraint is one of the most important factors in defining where you are at (e.g., equi-training-compute and equi-serving-compute already have huge ramifications). For tasks this common, it is also fairly reasonable for the authors themselves to present a sweep of best-effort experiments on various parameters, especially since many literatures are not really aiming for claiming SoTA (e.g. dictionary learning in interpretability research).

In abstract, you wrote "Surprisingly, ALICE generalizes to unseen ciphers after training on only ∼1500 unique ciphers, a minute fraction (3.7 × 10−24) of the possible cipher space.". Why is it surprising? Looking at it alphabet-by-alphabet, do the ciphers contain all the possible vocabulary mapping already, or you meant that the training set only has partial information and it generalizes to full vocabulary mapping?

"This task is combinatorially complex and requires reasoning over a space with V! possible ciphers ...". Why is it combinatorially complex? The way that presents it here seems extremely misleading. In the space of bijective maps between vocabularies, yes, we are talking about V! functions. For any language modeling tasks (toy tasks such as arithmetics, or real applications such as language translations), if we talk about the entire function space, we in most cases have orders of magnitude larger amount of functions than this.

Appendix I mentioned that LLM fails at this task. Yes, this is a very difficult task in a prompting-only regime. But have you tried supervised fine-tuning so that the methodology becomes apple-to-apple comparing with this paper?

---

> ### Author Response · Authors · 2025-11-28
>
> We thank the reviewer for taking the time to carefully assess our manuscript, and are glad to hear that the reviewer finds that we have "clear presentation and thoughtful ablations." Below we hope to clarify some of the points raised by the reviewer.
>
> ## Weaknesses
>
> > “Lack of novelty. There isn't much insight that is new”
>
> We refer the reviewer to our response to reviewer N9xb for a list of our novel contributions, expanded upon from their description in the paper.
>
> ## Questions
>
> > SOTA claim
>
> We thank the reviewer for pointing out these questions in regards to the SOTA claim. Our SOTA claim is intended to be within the scope of previously published cryptogram-solving systems in terms of performance, and accounting for serving-time compute cost. For many search-based baselines, there is no training cost, but inference time solving takes a very long time—part of the reason why neural network methods provide a promising alternative. Unfortunately, neural network approaches (specifically, Aldarrab and May (2021) and Kambhatla et al. (2023)) do not release code or provide sufficient implementation detail for exact reproduction. As a result, equi-training-compute comparison across all baselines is difficult to perform. However, in Appendix H we have reported inference-time compute cost as compared to baselines. In particular, the fastest previous model, Aldarrab and May (2021), decrypts at 400 char/s on an NVIDIA V100, whereas even on a CPU we decrypt 10x faster, and on an H100 we decrypt at 1.2M char/s. Adjusting for the difference in FLOPS between an H100 and V100 for FP32 calculations (~67 TFLOPS and ~16TFLOPS, respectively), the compute cost of our model at inference time is ~700x lower.
>
> > surprising generalization properties
>
> It is surprising that ALICE generalizes to unseen ciphers after only training on ~1500 unique ciphers, because these ciphers do *not* contain all the possible vocabulary mappings. Each sequence passed into the model comes from a sequence of data in our train split and is encrypted with one of the 1500 possible ciphers that have been generated in this 1500-cipher pool. We then test on a sequence of data from our test split, encrypted with a new, never-seen-before cipher from the 26! (10^26) possible ciphers (these are all the possible permutations of the English alphabet). So it is completely impossible for ALICE to have memorized a cipher that it saw in training, or even to memorize the training data and perform well on the held-out test data. Some training and test sequences will also not contain all the letters in the alphabet, so the model may not even see the entire mapping/cipher for that particular sequence. The combination of unseen data and unseen ciphers at test time is what makes it so surprising that ALICE can learn in general how to solve substitution ciphers despite seeing 1500 different ciphers applied on the training data. Thus, the model is generalizing to the full space of all possible ciphers (alphabet mappings), despite only seeing 1500 different examples of these ciphers (alphabet mappings).
>
> > combinatorial complexity
>
> The specific task that we tackle here (simple substitution ciphers) is combinatorial due to the nature of substitution ciphers. Each permutation of the alphabet created for the ciphertext encoding maps back to the original English alphabet used to create the plaintext. This means that for ‘A’ in the plaintext, there are 26 possible options for the corresponding ciphertext letter. For ‘B’, there are then 25 possible options, since the letter chosen for ‘A’ cannot be chosen again, for ‘C’ there are 24 options, and so on, leading to 26! possible mappings. So, the task space being the mapping space here means that the task/search/solution space is indeed combinatorial. We agree that other language modelling tasks may consist of a different number of functions; we wanted to clarify about the structure of the particular task that we are dealing with in this work.
>
> > supervised training of LLMs
>
> We thank the reviewer for the suggestion, and agree that fine-tuning an LLM for solving cryptograms would indeed be very interesting. For the purposes of this comparison, however, we simply wanted to see whether SOTA LLMs could solve cryptograms out of the box without any additional training. While we have not done supervised fine-tuning of such LLMs on this task, the backbone of the model in Kambhatla et al (2023), which we compare to in Table 2, follows a standard causal LM architecture and is trained in a supervised manner. Of course this is not pretrained on large corpora as a SOTA LLM might be, but neither is ALICE.

---

### Official Review · Reviewer_N9xb · 2025-10-31

**Soundness:** 4
**Presentation:** 2
**Contribution:** 1
**Rating:** 2
**Confidence:** 4

**Summary:**

Summary:
- Introduces cryptogram solving as a controlled testbed for studying reasoning and generalization in neural networks.
– Models must decrypt substitution ciphers without seeing the cipher key, facing a combinatorial search space of 26!.
– Proposes ALICE, a simple encoder-only Transformer that achieves state-of-the-art accuracy and speed on decryption tasks.
– Offers general architectural and analytic insights into model interpretability and generalization beyond cryptography.

**Strengths:**

I am a fan of using ciphers (and broadly encryptions) to better understand the nature of compression and intellihgent behavior in AI systems. So overall, broadly a fan of the direction.

**Weaknesses:**

The work motivates the study of cryptogram solving as a testbed for "understanding reasoning and generalization in neural networks". If I had seen this work 5 years ago, I would have been very excited. But at this point it is not clear to me what new knowledge it is adding to our current understanding.

I do not find the contributions as compelling. For example,  "novel decoding head that explicitly enforces bijectivity" sounds rather incremental: unclear if your work is adding on top of what we have known from Mena et al.



==A few related work==

in the context of modern LMs, there are works that use ciphers that should be discussed:
 - [1] https://arxiv.org/pdf/2504.19395
 - [2] https://arxiv.org/abs/2308.06463

There is also a classical work in NLP on ciphers. Few examples, among others:

 - [3] Kevin Knight, Anish Nair, Nishit Rathod, and Kenji Yamada. 2006. Unsupervised analysis for decipherment problems.
 - [4] Qing Dou and Kevin Knight. 2012. Large scale decipherment for out-of-domain machine translation.
 - [5] Nima Pourdamghani and Kevin Knight. 2017. Deciphering related languages.
 - [6] Sujith Ravi and Kevin Knight. 2008. Attacking decipherment problems optimally with low-order n-gram
models.
 - [7] Sujith Ravi and Kevin Knight. 2011. Bayesian inference for zodiac and other homophonic ciphers
 - [8] Taylor Berg-Kirkpatrick, Greg Durrett, and Dan Klein.
2013. Unsupervised transcription of historical documents.

**Questions:**

No questions at this point.
Looking forward to the authors' response for further exchange.

---

> ### Author Response · Authors · 2025-11-28
>
> We thank the reviewer for pointing out additional related work; we will be sure to reference these in our updated manuscript. We are glad to hear that the reviewer is "broadly a fan of the direction" and look forward to further discussion with the reviewer.
>
> Our main contributions are outlined below. **In summary**, we develop an architecture that significantly improves performance and eliminating structural errors in a way that has not been done for neural networks before **(1, 2)**, disprove the belief that standard embeddings do not work for solving ciphers **(3)**, and conduct novel analyses which show for the first time that neural networks use human-like strategies for solving cryptograms and can generalize well after even having seen only a small fraction of possible ciphers **(4, 5)**.
>
> 1. Our bijective decoding head is the first among neural-network based approaches
>  to explicitly enforce the structural constraints (bijectivity) of the substitution cipher problem. This is a gap between neural networks and, for example, search-based methods which have bijectivity built in. Our architectural component adds this advantage from these other methods and improves upon the inherent weakness in NN-based methods for this style of task.
>
> 2. We significantly improve upon the best performing neural decryption model for simple substitution ciphers, reducing errors from ~8% to ~1% on shorter sequences.
>
> 3. The current claim/belief in the field (e.g., in Aldarrab and May (2021)) is that because the meaning of a letter changes with different ciphers, learning embeddings for letters does not work and in fact leads to poor performance. This has motivated specialized encoding schemes such as the frequency encoding scheme in Aldarrab and May (2021) and the later recurrence encoding scheme in Kambhatla et al (2023). We *overturn this claim* and show that (surprisingly) a Transformer with a standard embedding scheme is capable of achieving strong performance, despite the meaning of letters changing with each example.
>
> 4. We provide insights into the inner workings of the model by, for example, performing early exit and probing experiments that produce a highly interpretable picture of the step-by-step process by which the model solves a cryptogram, and showing for *the first time* (to the best of our knowledge), that a neural network's reasoning process on this task is similar to that of a human (first matching letters, then matching words).
>
> 5. We have carried out the first study of the threshold for generalization in cryptogram solving by neural networks. We show that the number of unique ciphers that are required in order to achieve strong performance on unseen ciphers is surprisingly low; just ~1500 ciphers are needed in order to achieve near-perfect performance in a space with 26! possibilities. This is a novel finding that can provide insights into neural network generalization on other tasks as well.

---

### Note · Authors · 2026-01-04

I have read and agree with the venue's withdrawal policy on behalf of myself and my co-authors.